# Identification of Intron Retention in the Slc16a3 Gene Transcript Encoding the Transporter MCT4 in the Brain of Aged and Alzheimer-Disease Model (APPswePS1dE9) Mice

**DOI:** 10.3390/genes14101949

**Published:** 2023-10-17

**Authors:** Ayman EL-Seedy, Luc Pellerin, Guylène Page, Veronique Ladeveze

**Affiliations:** 1Laboratory of Cellular and Molecular Genetics, Department of Genetics, Alexandria University, Aflaton Street, El-Shatby, Alexandria 21545, Egypt; ayman.el-seedy@alexu.edu.eg; 2Neurovascular Unit and Cognitive Disorders (NEUVACOD), Faculty of Pharmacy (GP), Faculty of Fundamental and Applied Science (VL), University of Poitiers, Pôle Biologie Santé, 86073 Poitiers, France; guylene.page@univ-poitiers.fr; 3IRMETIST, INSERM, Faculty of Medicine, University of Poitiers (U1313), CHU de Poitiers, 86021 Poitiers, France; luc.pellerin@univ-poitiers.fr

**Keywords:** Slc16a3 gene, MCT4 transporter, intron retention, alternative splicing, astrocyte, Alzheimer, aging

## Abstract

The monocarboxylate transporter 4 (MCT4; Slc16a3) is expressed in the central nervous system, notably by astrocytes. It is implicated in lactate release and the regulation of glycolytic flux. Whether its expression varies during normal and/or pathological aging is unclear. As the presence of its mature transcript in the brain of young and old mice was determined, an unexpectedly longer RT-PCR fragment was detected in the mouse frontal cortex and hippocampus at 12 vs. 3 months of age. Cultured astrocytes expressed the expected 516 base pair (bp) fragment but treatment with IL-1β to mimic inflammation as can occur during aging led to the additional expression of a 928 bp fragment like that seen in aged mice. In contrast, cultured pericytes (a component of the blood–brain barrier) only exhibited the 516 bp fragment. Intriguingly, cultured endothelial cells constitutively expressed both fragments. When RT-PCR was performed on brain subregions of an Alzheimer mouse model (APPswePS1dE9), no fragment was detected at 3 months, while only the 928 bp fragment was present at 12 months. Sequencing of MCT4 RT-PCR products revealed the presence of a remaining intron between exon 2 and 3, giving rise to the longer fragment detected by RT-PCR. These results unravel the existence of intron retention for the MCT4 gene in the central nervous system. Such alternative splicing appears to increase with age in the brain and might be prominent in neurodegenerative diseases such as Alzheimer’s disease. Hence, further studies in vitro and in vivo of intron 2 retention in the Slc16a3 gene transcript are required for adequate characterization concerning the biological roles of Slc16a3 isoforms in the context of aging and Alzheimer’s disease pathology.

## 1. Introduction

The SLC16A gene family encodes related solute carriers and is composed of 14 transmembrane proteins. These transporters have been identified in all eukaryotic organisms of which genomes have been sequenced to date. In mammals, four members of this family have been characterized as monocarboxylate transporters. MCT1, MCT2, MCT3 and MCT4 are responsible for the proton-linked transport of several monocarboxylates, such as pyruvate, lactate and ketone bodies across the plasma membrane [1]. MCT4 has been reported as the transporter with the lowest affinity but it exhibits a high capacity for lactate transport [2]. For this reason, its expression is associated with tissues and cell types which exhibit high glycolytic activity. This is the case in fast-twitch muscles, in which it mediates lactate efflux [3]. In the central nervous system, MCT4 is expressed prominently by astrocytes [4] but it was also reported in endothelial cells [5] as well as in microglia [6]. Due to the presence of a hypoxia responsive element (HRE) in its promoter, MCT4 expression in astrocytes has been demonstrated to be regulated by oxygen levels under the control of a hypoxia-inducible factor-1 (HIF-1)-dependent pathway [7]. MCT4 expression was also shown to be prominent under certain pathological conditions. This is the case in cancer, in which MCT4 plays a major role in glycolytic tumor cells [8]. In the central nervous system, MCT4 expression was shown to be modified following a stroke, with an appearance in neurons which do not normally express this transporter [9].

Changes in brain aerobic glycolysis have been described in both aging [10] and Alzheimer’s disease (AD) [11,12]. However, relatively few studies have investigated the possible changes in brain MCT4 expression that might occur during normal and/or pathological aging. A first study reported a decrease in MCT4 expression in the brain of an APP/PS1 mouse model for AD at the age of 3 months old [13]. This reduction can impede lactate transport and could induce lactate-deficient neurons and thus worsen energy deficiency in neurons. In contrast, another study described an upregulation of hippocampal MCT4 expression in the same APP/PS1 mouse model for AD at the age of 2 and 3 months old [14], suggesting an essential role for MCT4. Indeed, after knockdown expression of MCT4 in the hippocampus of AD model mice, the cognitive ability of mice was improved and the apoptosis rate of hippocampal neurons was significantly reduced. In order to obtain further insight about possible mechanisms that might explain variations in MCT4 expression between young and senescent animals, we performed RT-PCR in the frontal cortex and hippocampus of wildtype mice and subsequently in the APP/PS1 mouse model of Alzheimer’s disease. Our results led us to identify an alternative splicing event for the MCT4 transcript with a specific intron retention that seems to be associated with both normal and pathological aging. Moreover, it seems that such an alternative splicing event can be triggered in cultured astrocytes by exposure to an inflammatory signal like IL-1β. It is postulated that intron retention for the MCT4 transcript might be a cellular stress signal in the central nervous system associated with normal and pathological aging.

## 2. Materials and Methods

### 2.1. Animals

B6C3F1 WT (wild type) and transgenic APPswePS1dE9 mice were purchased from Mutant Mouse Resources and Research Centers (Stock No: 34829-JAX, Bar Harbor, ME, USA). The transgenic APPswePS1dE9 model displayed the Alzheimer phenotype (authorization from “Haut Comité de Biotechnologie français” (HCB) to Pr Guylène Page, number 2040 for reproduction, treatment, behavioral tests and ex vivo experiments). WT and transgenic APPswePS1dE9 were obtained by crossing a male APPswePS1dE9 mouse with a WT female mouse, as explained previously [15]. All animal care and experimental procedures conformed to the French Decret number 2013–118, 1 February 2013 NOR: AGRG1231951D in accordance with European Community guidelines (directive 2010/63/UE). The use of animals was approved by the Ethical and Animal Care Committee (N°84 COMETHEA, Ethical Committee for Animal Experimentation Poitou-Charentes, France). At weaning, all mice were genotyped by polymerase chain reaction (PCR) analysis of tail biopsies according to the manufacturer’s recommended protocols (CliniSciences, Nanterre, France). All efforts were made to minimize animal suffering, as well as the number of animals used. The animals were housed in a conventional state under adequate temperature (23 ± 3 °C) and relative humidity (55 ± 5%) control, with a 12/12 h reversed light/dark with access to food and water ad libitum.

### 2.2. Cell Cultures

The WT cell line of astrocytes was developed in the NEUVACOD laboratory by an immortalization process of primary brain astrocytes (data not published). They were cultured in a DMEM 4.5 g/L glucose medium, supplemented with 10% (*v*:*v*) of certified fetal bovine serum (Fisher scientific, Illkirch, France), 1% of 2 mM glutamine, 1% of 50 μg Streptomycin/mL and 50 U Penicillin/mL mixture at 37 °C with 95% air: 5% CO_2_ atmosphere. Cell lines of endothelial cells and pericytes were obtained from mouse primary cultures according to a protocol described in a patent (FR17/57643 and WO/2019/030380). All these cell lines were obtained from mouse brain at 3 months old.

### 2.3. Lipopolysaccharide and Interleukin-1β Treatments

Mouse endothelial cells, astrocytes and pericytes cells were plated at a density of 0.3 × 10^6^ cells/mL in 6-well plates. These three types of mouse brain cell lines were treated with the indicated concentrations of LPS (100 ng/mL, Sigma-Aldrich, L2887-10MG, Saint Louis, MO, USA) or IL-1β (400 pg/mL, Biolegend, San Diego, CA, USA) for 48 h, as described in [16]. The vehicle control group received the same volume of deionized water at the same time. Each culture was performed in triplicate. After the incubation period, cell suspensions were prepared and used for RNA isolation and further analysis.

### 2.4. RNA Isolation

Firstly, mouse endothelial cells and pericytes were detached with a solution of TrypLE (Gibco, Warwickshire, UK). After cell detachment, medium was added and cells transferred to a 50 mL Falcon tube, then centrifugated at 1500 RPM for 5 min to form a pellet. Total RNA was isolated from cell pellets after removing the supernatant using the RNeasy kit (Qiagen, Hilden, Germany), according to the manufacturer’s instructions. In our experiments, four WT mice of each age (3-month, 6- and 12-month-old) were used as 3-, 6- and 12-month-old AD mice. WT and AD mice were euthanized, then the frontal cortex (CF) and the hippocampus (HI) were collected in tubes containing RNA lysis buffer. Total RNA was extracted from the cortex and the hippocampus by using the SV total RNA isolation kit (Promega, Madison, WI, USA) and subjected to digestion with 1U RNase-free DNase I treatment before RNA elution, as supplied with the SV RNA isolation kit and following the manufacturer’s protocol. RNA concentration was measured using the NanoDrop 2000 Spectrophotometer (Thermo Scientific, Wilmington, DE, USA) in all brain samples.

### 2.5. RT-PCR Amplification

One µg of total RNA was reverse transcribed to cDNA using the ImProm-II™ Reverse Transcription kit (Promega, Charbonnieres, France). cDNA was amplified by polymerase chain reaction (PCR) to determine expression of SLC16A3 gene encoding the MCT4 transporter in three types of mouse brain cell lines (in vitro model) and in the brain of WT and transgenic APPswePS1dE9 mice (in vivo model). The RT-PCR was accomplished with targeted primers: MCT4-F, 5′-TATCCAGATCTACCTCACCAC-3′ (forward) and MCT4-R, 5′GCCTGGCAAAGATGTCGATGA3′ (reverse). SLC2A1 (Glut1) was used as internal control gene for MCT4 amplification. The primer sequence was F-5′-CTTCATTGTGGGCATGTGCTTC-3′, R-5′-AGGTTCGGCCTTTGGTCTCAG-3′. The PCR was conducted using GoTaq**^®^** Green Master Mix (Promega, Madison, USA). The program used was 5 min at 95 °C for initial denaturation, followed by 35 cycles of 95 °C (30 s, denaturation), 60 °C (30 s, annealing), and 72 °C (45 s, elongation), and 5 min at 72 °C for the final elongation. The sizes of RT-PCR products were 516 bp (mature transcript, absence of intron 2 retention) and 928 bp (immature transcript, the presence of intron 2) which was electrophoresed on a 1.5% agarose gel stained with ethidium bromide (0.5 mg/mL). Ladder: ØX174DNA/*Bsu*RI (*Hae*III) (Thermo Fisher Scientific, Swindon, UK) was used as a molecular weight reference. Gel images were captured using G: BOX F3 Gel Imaging System (Syngene, Cambridge, UK). RT-PCR product from the MCT4 constitutive transcript in endothelial cells treated or not treated were quantified using GeneTools analysis software 4.3.17 (Syngene, UK). All samples were quantified and were normalized to Glut1 transcript from three independent experiments.

### 2.6. DNA Sequencing of RT-PCR Products

Sequencing was performed bi-directionally using the BigDye X-Terminator purification kit (Applied Biosystem, USA). RT-PCR products were analyzed on a 3130 Genetic Analyzer (Applied Biosystems). The sequence obtained was compared with the reference cDNA sequence from GenBank, the Ensembl Genome Browser 91, (https://www.ensembl.org/index.html; accessed on 7 August 2023), against reference mouse gene sequences from (Mct4 >ENSMUST00000070653.13 Slc16a3-201 cdna: protein_coding) and appropriately examined to identify regions of sequence homology.

### 2.7. Statistical Analysis

All data for the differential expression of MCT4 constitutive transcript in endothelial cells with or without LPS and IL-β treatments (three independent cultures in each case) were analyzed using GraphPad Prism 5 (GraphPad Software, San Diego, CA, USA). The findings were analyzed using one-way analysis of variance to detect differences between cells with LPS and IL-β treatments and cells without treatment as a control. A Kruskal–Wallis test with a Dunns multiple comparison test (a post hoc test for the Kruskal–Wallis test) was applied for multiple variable comparisons. The significance level was set at *p* < 0.05.

## 3. Results

### 3.1. Detection of Intron Retention in Aged Brain MCT4 mRNA

The expression profile of the Slc16a3 transcript coding for the transporter MCT4 was evaluated in vivo by RT-PCR in the frontal cortex and hippocampus of wildtype (WT) mice at 3, 6 and 12 months old. A normal splicing was detected in 3-month-old WT mice for both brain regions (cortex frontal and hippocampus) with a fragment at 516 bp (Figure 1A). However, in 12-month-old mice, RT-PCR products of the Slc16a3 gene revealed a unique fragment of 928 bp in the cortex and hippocampus (Figure 1B).

In order to attempt to identify the cellular origin of these different fragments, we analyzed the expression profile in different cultured brain cells. RT-PCR analysis of the Slc16a3 gene in mouse astrocytes (ACs, known to normally express MCT4) detected a single fragment of 516 base pair (bp) corresponding to normal splicing (Figure 1C) without treatment and with treatment by LPS. However, both fragments were observed in astrocytes treated with IL-1β (Figure 1C), despite the fact that the 928 bp fragment was expressed at a lower level (21%) and most of the MCT4 mRNA (79%) was of normal size. RT-PCR analysis of the Slc16a3 gene in mouse pericytes (PCs, cells that are part of the blood–brain barrier with astrocytes) showed a single fragment of 516 base pair (bp) corresponding to normal splicing. Cultured pericytes with either LPS or IL-β treatment only expressed the 516 bp fragment (Figure 1D). MCT4 gene analysis by RT-sqPCR revealed differential expression in endothelial cells with or without LPS and IL-1β treatments, representing two mRNA forms: one corresponding to normal splicing with a 516 bp sequence and another one at the 928 bp fragment (Figure 1(E,1)). After semi-quantifying RT-PCR products for the constitutive exon, we observed that both treatments reduce MCT4 mRNA expression, but it became significant only following the treatment with IL-1β in comparison to non-treated cells (Figure 1(E,2)). No difference in the levels of Glut1, an internal control, was observed among cells with or without treatment.

RT-PCR products were bi-directionally sequenced on a 3130 Genetic Analyzer (Applied Biosystems) and all results were confirmed. Chromatograms revealed two different sequences (Figure 1F,G) accounting for the 516 and 928 bp fragments. After extraction of the highest band, a sequence containing a partial exon 2, the full intron 2 and a partial exon 3 was identified without any detection of genetic variants, even in the intron sequence. This sequence was deposited in Genbank and the accession number is BankIt2672483Slc OQ459712.

### 3.2. A Proposed Mechanism of Alternative RNA Splicing for the MCT4 Transporter in Aged Mice

Based on the sequence analysis of both fragments, a mechanism of alternative RNA splicing is proposed to occur between young and aged wildtype mice. In young mice, normal splicing will give rise to the classical MCT4 protein production (Figure 2A). In aged mice, aberrant splicing with intron 2 retention (Figure 2B) inducing a premature termination codons (PTC) site seems to take place, with the possibility of absent or truncated MCT4 protein production. The presence of intronic PTC suggests that this RNA form could be degraded by a nonsense-mediated (NMD) RNA decay mechanism (Figure 2B). However, there is also a possibility of splicing to remove intron 2 and obtain the mature transcript and produce the full protein (Figure 2C). As shown in Figure 2C, the predicted amino acid (aa) sequence of the normal protein is 470aa in length according to Gene bank database, with a mass (Da) of 50,373 (https://www.ensembl.org/index.html; accessed on 7 August 2023). In Figure 2D the putative truncated protein (128aa) that could be arising from abnormal splicing is illustrated. The alignment of both aa sequences was realized to detect their similarities and variations (Figure 2E), but, apart from the PTC, no polymorphism was detected.

### 3.3. Detection of Intron Retention in an Alzheimer Mouse Model

To determine whether in age-related neurodegenerative diseases such as AD intron retention was also detected, expression profiling of the MCT4 transporter was performed in vivo by RT-PCR in AD mouse brain. RT-PCR products of the Slc16a3 gene revealed a unique fragment of 928 bp in the cortex and hippocampus of 12-month-old AD mouse brain (Figure 3A), suggesting a similar intron retention as detected in wildtype mice. Chromatograms confirmed the same abnormal sequences containing partial exon 2, full intron 2 and partial exon 3, like that obtained in wildtype mice (Figure 1G) without any detection of genetic variants. Quite surprisingly, and in contrast with the situation in 3-month-old wildtype mice, the 516 bp fragment, as well as the 928 bp fragment, were absent in the cortex and hippocampus of 3-month-old AD mouse brain (Figure 3B). As a positive control, Glut1 was used and since its detection was clear, it can be assumed that RNA extraction and cDNA production have been correctly performed (Figure 3C).

### 3.4. Detection of Non Intron Retention in 6-Month-Old Mice

To determine when intron retention could be detected, expression profiling of the MCT4 transporter was performed in vivo by RT-PCR in WT and AD mouse brain at an intermediate age between 3 months and 12 months, namely 6 months old. After 3 months old with AD progression, the expression of MCT4 will be expressed correctly, as we found in 6-month-old mice (WT and AD model) but with the decrease in the mature form of the MCT4 transcript (Figure 3D,E) until 12 months old, where immature transcripts were presented.

## 4. Discussion

Alternative splicing (AS) is a regulatory mechanism of gene expression that allows the expression of several transcripts (mRNAs) from the same gene sequence. AS has contributed to the widespread diversification of the proteome across eukaryotic taxa [17,18]. It can be divided into seven types: cassette exons (exon skipping), mutually exclusive exons, alternative 5′ splice sites, alternative 3′ splice sites, alternative 3′ terminal exons, alternative 5′ exons and intron retention (IR) [19,20]. Among these mechanisms, intron retention (IR) is a form of AS characterized by the inclusion of intronic sequence in a mature mRNA transcript. If exon skipping occurs in 40% of superior eukaryotes, IR was considered until recently as nearly absent and/or irrelevant in animals [21,22]. It is important to consider that many alternative splicing events (ASE) are regulated to produce specific protein isoforms [23,24]. Of these different classes of ASE, IR was not particularly identified as an important mechanism of gene expression regulation in humans and other mammals, at least until recently. This may have resulted from the difficulty in determining clearly an apparent IR event.

Indeed, detection of IR may seem surprising, as mechanisms exist to eliminate such aberrant transcripts. Nonsense-mediated mRNA decay (NMD) is one of several cytoplasmic surveillance mechanisms that monitor mRNA translation in eukaryotes, targeting mRNAs containing a premature termination codon (PTC) for rapid degradation [25,26,27,28]. However, Ameur and colleagues [29] observed significant numbers of sequencing reads mapping to introns in total RNA libraries from human brain cells. Furthermore, many human cancers seem to exhibit intron retention [30]. The original view was that NMD’s function was confined to quality control. Removal of mRNAs from genes carrying nonsense mutations avoids the excess of potential truncated polypeptides [31,32]. In fact, IR is often related to down-regulation of gene expression via NMD (IR-NMD) [33]. However, an increase in IR was discovered recently to be a common feature in age-related genome-wide splicing analyses. For example, in Saccharomyces cerevisiae, in Caenorhabditis elegans, in Drosophila melanogaster, in mus musculus (mouse), and even in humans, the prevalence of IR seems to increase with aging [34,35]. Alzheimer’s disease (AD) can be considered as an accelerated process of senescence. Interestingly, the occurrence of IR was identified in up to ~780 genes in AD brain tissue [36]. Genes susceptible to IR are particularly enriched in pathways related to the regulation of mRNA and protein homeostasis, but include other key proteins for brain function. This the case for the protein Tau, which is responsible for the formation of neurofibrillary tangles observed in AD [37].

A recent study detected the presence of alternative RNA splicing for the MCT4 transcript in melanoma, basal-cell carcinoma and squamous-cell carcinoma tumors in Greek patients [38]. These authors identified twelve miRNAs produced from intron retention of the MCT4 transcript. Moreover, they suggested the putative synthesis of MCT4/SLC16A3-216 intron-derived novel proteins: this sequence accommodates the AC132872.4-202 pseudogene, overlaps with the CSNK1D-204 gene, and can be transcribed from the forward and from the reverse strand (CSNK1D-204). As the retained mouse intron contains only 412bp versus 1376bp in the human sequence, miRNAs seem not be produced from this intron, and no other sequence homology was detected using the blast program. Interestingly, it was observed that the same MCT4 intron 2/3 is retained in brain samples from aged or Alzheimer’s disease mice, as in non-melanoma and melanoma biopsy specimens from Greek patients [38]. Thus, it suggests that a common IR mechanism exists to regulate MCT4 expression in different tissues and cell types as well as distinct pathological conditions. In fact, the prevalence of IR in multiple human as well as mouse cells and tissues is more important than previously thought [39]. IR plays important roles in mammalian cells during cellular differentiation, cell activation, stress or cancer [40]. Indeed, prevalent IR could modulate gene expression and contribute to transcriptome diversity with senescence**-**associated phenotypes, largely extending the biological significance of IR [41].

A striking switch from the mature MCT4 transcript in young mice to complete IR in older mice occurs during normal aging. It would be interesting to determine more precisely at which precise age it occurs and whether there is a transition period during which both transcripts are present. Moreover, it would be important to examine whether this alternative splicing event has any impact on the expression and localization of the MCT4 protein. It was shown that regulated IR in fully transcribed RNAs represents a mechanism for rapidly mobilizing a pool of mRNAs in response to neuronal activity [42]. Indeed, transcripts with introns are substantially expressed and stored in the nucleus. Upon stimulation, these transcripts undergo rapid intron excision, and fully spliced mRNAs are exported to the cytoplasm to undergo translation. Such a mechanism could operate in aged mice to better respond to situations requiring enhanced glycolysis and lactate release.

Quite intriguingly, while only the MCT4 transcript with intron retention is detected in the frontal cortex and hippocampus of 12-month-old AD mice, no MCT4 transcript was identified in brain tissues from 3-month-old AD mice and only normal transcripts in 6-month-old mice (WT and AD). Cellular lactic acid secretion is thought to be predominantly mediated by MCT4, a plasma membrane transporter protein [43]. MCT4 is involved in energy metabolism during early pathological processes in AD. Thus, the suppression of MCT4 expression in early AD pathology (3-month-old mice) could be a neuroprotection against this stage of AD disease [14], which impairs lactate transport from astrocytes to neurons [44]. However, at the age of 6months old a decrease in normal transcript without the detection yet of aberrant splicing was observed, suggesting a decrease in this neuroprotection.

One possible interpretation is that intron retention for the MCT4 gene occurs already in young AD mice but the NMD mechanism is correctly operating and successfully eliminates what is considered an aberrant transcript. It would be interesting to determine in brain samples from this AD mouse model at which age intron retention for the MCT4 transcript can start to be detected. If we consider the evolution of the MCT4 transcript with age, based on our observations, we can hypothesize the occurrence of distinct phases. First, in young animals, only the normal transcript would be produced and detected. Then, as IR starts to take place with the animal ageing, the NMD mechanism would entirely remove this aberrant form with intron retention. When the normal transcript is no longer produced and NMD eliminates all transcripts with intron retention, no transcript would be detected. As the NMD mechanism starts to fail, the transcript with intron retention would start to be detected. The same sequence of events might take place in both WT and AD animals, but with a different time course. It could explain why observations differ in three-month-old animals for WT and AD. If confirmed, early detection of intron 2 retention for the MCT4 transcript (and/or the lack of MCT4 transcript detection) could become useful as an early biomarker of AD.

In older animals, the same mechanism might not be as efficient and allows the accumulation of a significant fraction of this transcript, explaining its detection. Indeed, increased intron retention has been proposed as a pathogenic mechanism in both ageing and Alzheimer’s disease [34]. It would be interesting to determine in brain samples from this AD mouse model at which age intron retention for the MCT4 transcript can be detected. Thus, detection of early intron 2 retention for the MCT4 transcript (and/or the lack of MCT4 transcript detection) could become useful as showing early biomarkers of AD.

In order to obtain an insight into the cellular origin of MCT4 intron retention, investigations were performed in three brain cultured cell types. First, both MCT4 transcripts, with and without intron retention, were detected in immortalized mouse brain endothelial cells. Expression of MCT4 in endothelial cells has been reported previously [5]. Whether the intron retention mechanism observed here might play a specific role in the regulation of MCT4 expression in this cell type will require further investigation, although treatment with inflammatory signals does not exert any effect. Pericytes are an important component of the blood–brain barrier surrounding blood vessels. So far, no monocarboxylate transporters have been reported to be expressed by these cells. Our detection of the mature MCT4 transcript suggest they might express the MCT4 protein. However, treatment with inflammatory signals does not appear to cause intron retention of MCT4 transcript like that in astrocytes. Finally, astrocytes were known to express MCT4 both in vitro and in vivo [7,9]. Here it could be confirmed that cultured astrocytes express the mature MCT4 transcript. Interestingly, treatment of cultured astrocytes with IL-1β (but not LPS) to mimic inflammation led to the detection of intron retention for the MCT4 transcript. The absence of alterations in MCT4 gene expression for astrocytes treated with LPS might be due to the fact that this activator is not sufficient alone to induce inflammation in these cells [45,46]. Moreover, LPS exposure did not alter JNK and phosphorylated (p)-JNK levels after TLR4 activation in astrocytes or neurons in co-cultures [47]. It significantly increased p38 and p-p38 levels in neurons in co-cultures, whereas it only increased p-p38 in astrocytes [47]. Importantly, in vitro JNK and p38 are activated in astrocytes stimulated by proinflammatory cytokines like IL-1β, and inhibition of JNK and p38 reduces expression of inflammatory endpoints like inducible NO synthase (iNOS), TNFα, and IL-6 [48]. In addition, Guo et al. [49] revealed that the ASK1-p38 axis is required for chemokine production in astrocytes through activation of multiple TLRs. Thus, the appearance of intron retention for the MCT4 transcript in astrocytes triggered by inflammatory signals could be a sign of cellular stress and contribute to its detection with normal and pathological aging.

## 5. Conclusions

Our results demonstrate the existence of an intron retention mechanism for the MCT4 transcript in the brain. The prevalence of this phenomenon increases with age and is present in a neurodegenerative disease mouse model. Astrocytes seem particularly prone to intron retention of the MCT4 transcript, as it can be induced by treatment with a cytokine. As genetic variants even in the intron sequence could not be detected, IR is most likely obtained by the involvement of the spliceosome and not the implication of polymorphisms or mutations. Moreover, it is noteworthy and should be emphasized that both in vitro and in vivo experiments have unraveled the same splicing defect in the MCT4 transcript. Further studies with anti-NMD drugs in astrocytes treated with or without LPS or IL-1β could provide a proof of the possible role of NMD in eliminating the undetected mRNAs in 3-month-old AD mice and will be important for understanding its role in the regulation of MCT4 expression. Additionally, in vitro and in vivo characterization of the effects of intron 2 retention for the Slc16a3 gene transcript encoding the transporter MCT4 on the protein processing and immunolocalization will be necessary to confirm if the aberrant transcript will be translated to a truncated and/or non-functional protein.

## Figures and Tables

**Figure 1 genes-14-01949-f001:**
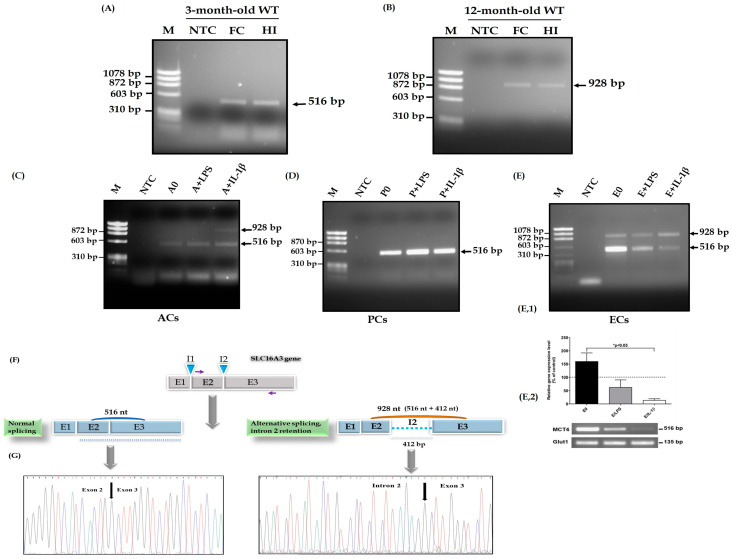
Gene expression profile of the Slc16a3 gene in mouse brain and in cultured brain cells. RT-PCR of the Slc16a3 gene in the frontal cortex (FC) and hippocampus (HI) of (**A**) 3-month-old or (**B**) 12-month-old wildtype mice (WT). RT-PCR of the Slc16a3 gene in (**C**) cultured astrocytes (ACs) treated or not with LPS or IL-1 β, (**D**) cultured pericytes (PCs) treated or not with LPS or IL-1β, (**E**) cultured endothelial cells (ECs) treated or not with LPS or IL-1 β representing two transcripts of Slc16a3 gene (**E,1**). Histograms representing the level of expression of the 516 bp fragment for MCT4 normalized to the level of expression of the 135 bp fragment for Glut1 in endothelial cells treated or not with LPS or IL-1β (**E,2**). Quantification of RT-PCR signaling revealed a decrease in MCT4 expression in LPS or IL-1β treatments (62.4% and 13.68%, respectively) compared to ECs alone (159.8%), N = 3. Sequence of the normal and alternative splicing with intron retention of the Slc16a3 gene giving rise to the 516 bp and the 928 bp fragments (**F**), and the associated chromatograms (**G**). NTC: non-template control. The number of mice for each experimental group is four.

**Figure 2 genes-14-01949-f002:**
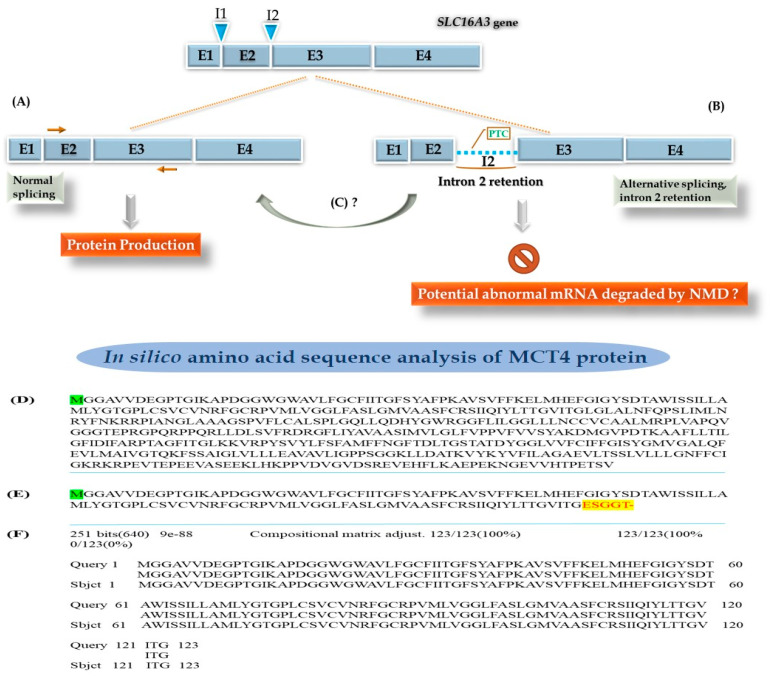
Alternative splicing of the Slc16a3 gene giving rise to intron retention. (**A**) Normal splicing which leads to the synthesis of the MCT4 protein. (**B**) Alternative splicing with intron retention. As intron 2 contains a premature termination codon (PTC), it would normally cause a degradation of this mRNA by a nonsense-mediated RNA decay (NMD) mechanism with no MCT4 protein synthesis, or alternatively the production of a truncated MCT4 protein. (**C**) Possible splicing to remove intron 2 and still obtain the mature transcript followed by the production of the full protein. (**D**) In silico amino acid sequence analysis demonstrated amino acid sequences of the normal MCT4 protein and the putative truncated form arising from the alternative splicing. Amino acid sequence of the normal protein (470aa) according to Gene bank, molecular weight of 50,373 Da. (**E**) Amino acid sequence of the predicted truncated protein (128aa) due to abnormal splicing. The yellow part displays the frame shift-enhancing amino acid modification. The dash indicates the end of translation. (**F**) Alignment of both amino acid sequences to highlight the absence of difference on the first 123 amino acids.

**Figure 3 genes-14-01949-f003:**
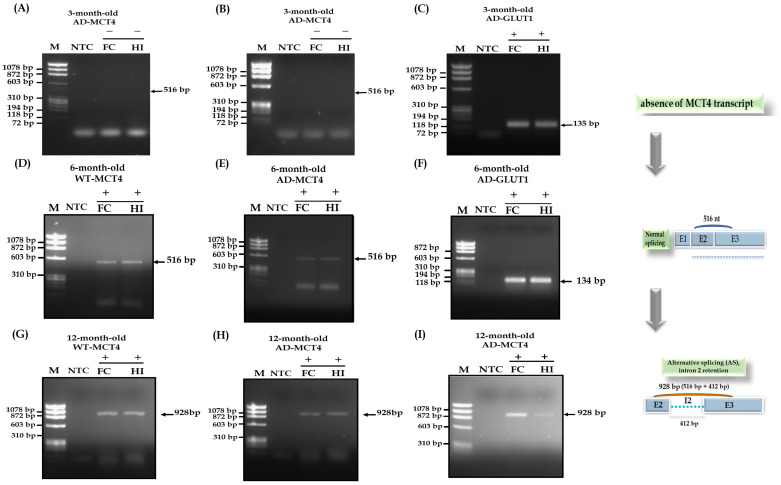
RT-PCR for the Slc16a3 gene in an Alzheimer mouse model. (**A**,**B**) RT-PCR for the Slc16a3 gene in the frontal cortex (**C**,**F**) and hippocampus (**H**,**I**) of 3-month-old Alzheimer (**A**,**D**) mice. (**D**,**E**) RT-PCR for the Slc16a3 gene in the frontal cortex (**C**,**F**) and hippocampus (**H**,**I**) of 6-month-old Alzheimer (**A**,**D**) mice. (**G**) RT-PCR for the Slc16a3 gene in the frontal cortex (**C**,**F**) and hippocampus (**H**,**I**) of 12-month-old WT mice. (**H**,**I**) RT-PCR for the Slc16a3 gene in the frontal cortex (**C**,**F**) and hippocampus (**H**,**I**) of 12-month-old Alzheimer (**A**,**D**) mice. A normal fragment is expected at 516 bp, and alternative splicing with intron retention would give rise rather to a fragment at 928 bp. (**C**,**F**) As control, RT-sqPCR for the Slc2a1 gene (coding for GLUT1) was performed and exhibited the expected 135 bp fragment (Appendix A). The number of mice for each experimental group is four.

## Data Availability

Currently, all presented data are available on request from the corresponding author.

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
