# Peer review of "Identification of Intron Retention in the Slc16a3 Gene Transcript Encoding the Transporter MCT4 in the Brain of Aged and Alzheimer-Disease Model (APPswePS1dE9) Mice"

_genes, 2023, doi:10.3390/genes14101949_

Round 1
Reviewer 1 Report
Summary: MCT4, expressed in the central nervous system by astrocytes, plays a crucial role in regulating monocarboxylic acid flux, particularly lactate. EL-Seedy et al. report intron retention in the MCT4 transcript in both aged mouse models and cultured cells. Intriguingly, this intron retention phenomenon is more pronounced in 12-month-old wildtype and Alzheimer's mouse models, as well as in astrocytes subjected to IL-1β, mimicking age-related inflammation. Detailed sequence analysis of the unique 928 bp PCR product reveals the presence of a partial exon 2, complete intron 2, and a partial exon 3. Based on these findings, the authors suggest that aberrant splicing leads to intron 2 retention, subsequently introducing a premature termination codon, ultimately resulting in the absence or reduction of MCT4 protein. While EL-Seedy et al.'s work is intriguing, it warrants additional experimental evidence to substantiate the claims made in the manuscript. Publication consideration could be given once the authors address the comments and concerns outlined below:
Major comments: The major comments are as follows:
1. In the last paragraph of the introduction (lines 57-60), the authors draw attention to an inconsistency in earlier research findings. Specifically, one study documented a reduction in MCT4 expression in brain samples from the APP/PS1 mouse model of Alzheimer's disease, while another study conducted on the same model reported an increase in MCT4 expression specifically in the hippocampus. To enhance the clarity of this inconsistency, it is recommended that the authors provide a more detailed explanation.Furthermore, it is important to note that the authors have not included critical information regarding the age of the mice used in these studies. We strongly recommend that this essential information be incorporated into the revised manuscript.
2. In Results Section 3.1, the authors assessed MCT4 transcript levels in 3- and 12-month-old WT mice. Incorporating data from 6-month-old mice would be valuable in determining the onset age of intron retention in the MCT4 transcript.
3. The rationale behind the authors' choice to treat cells with IL-1β for 48 hours before assessing MCT4 transcript levels in cultured astrocytes, pericytes, and endothelial cells is not clearly explained. Have the authors considered experimenting with a 24-hour time point? Clarification on this decision would enhance the understanding of the experimental design.
4. It seems there may be a correlation between the decrease in transcript levels of the 516 bp fragment and the appearance of the 928 bp fragment in IL-1β-treated astrocytes and endothelial cells. To substantiate these observations in Fig. 1C and E1, we recommend conducting densitometric analysis on the gel data.
5. It's worth considering why the authors haven't validated their results by assessing MCT4 protein levels. We suggest that the authors support their RT-PCR data by performing Western blot analysis to examine MCT4 protein levels in 3-, 6-, and 12-month-old WT mice and in cultured astrocytes, pericytes, and endothelial cells treated with or without IL-1β. This would help assess whether the protein levels of MCT4 align with the RT-PCR findings.
6. In the discussion section, the authors provide a speculative interpretation of their findings, suggesting that intron retention for the MCT4 gene might occur in young AD mice, but the NMD mechanism effectively eliminates what is considered an aberrant transcript. However, this interpretation would be more robust after examining the protein levels of MCT4 in all the models used in this study.
7. In the materials and methods section (2.4, lines 111-113), it is mentioned that "two WT mice of each age (3-month and 12-month old) were used as 12-month old AD mice whereas four 3-month old AD mice were tested." This statement lacks clarity and coherence. We recommend revising this section to explicitly specify the number of mice used in each experimental group.
8. To strengthen the validity of their findings, the authors should consider expanding the sample size. Instead of testing only 2 wild-type mice for the 3- and 12-month time points, we suggest including at least 8 mice for each of the 3-, 6-, and 12-month time points.
9. It is unclear how many mice were included in each group. Assuming there were multiple mice, it's puzzling why data from only one mouse is displayed in Fig. 1A, 1B, and Fig. 3. To enhance the robustness of the study, we recommend using a larger sample size, including a minimum of 8 mice per group (WT/AD – 3, 6, and 12 months), and running all the samples from each group on a single gel for validation purposes.
10. The description and details regarding the statistical analysis are insufficient and unclear. Please provide a more comprehensive and well-defined explanation of the statistical methods used in the study.
Minor comments: The minor comments are as follows:
1. In line 99, there is a typo: "0.3 x 10^6 cells/ml." The exponent "6" should be in superscript form (10^6).
2. In line 50, there is a typographical error in the phrase "This the case in."
It is advisable to conduct a comprehensive review of the manuscript's language and grammar to ensure accuracy and clarity throughout the document.
Author Response
We highly appreciate the reviewers’ insightful and helpful comments on our manuscript.
’
Reviewer 1
We tried to respond to the majority of critics.
Point 1: in the revised manuscript, we provided more details and specify the age of mice used in these studies.
“A first study reported a decrease of MCT4 expression in the brain of an APP/PS1 mouse model for AD at the age of 3 month-old [13]. This reduction can impede lactate transport and could induce lactate deficient neurons and thus worsen energy deficiency in neurons. In contrast, another study described an upregulation of hippocampal MCT4 expression in the same APP/PS1 mouse model for AD at the age of 2 and 3-month-old [14], suggesting an essential role of MCT4. Indeed, after knocking down expression of MCT4 in the hippocampus of AD model mice, the cognitive ability of mice was improved and the apoptosis rate of hippocampal neurons was significantly reduced. “
Point 2. In results section 3.1 we incorporated data from 6 month-old mice, since we were able to do these experiments in early September and we had the results recently. We added these results in a Figure 3 revised (3D 3E 3F).
Point 3. The rationale behind the authors' choice to treat cells with IL-1β for 48 hours before assessing MCT4 transcript levels in cultured astrocytes, pericytes, and endothelial cells is not clearly explained. Have the authors considered experimenting with a 24-hour time point?
These three types of mouse brain cell lines were treated with the indicated concentrations of LPS (100ng/mL, Sigma-AldrichL2887-10MG) or IL-1β (400pg/mL, Biolegend) for 48 hours, we used this timing as described in [16].
[16] François, A.; Terro, F.; Janet, T.; Rioux Bilan, A.; Paccalin, M.; Page, G. Involvement of interleukin-1β in the autophagic process of microglia: relevance to Alzheimer's disease. J Neuroinflammation. 2013;10:151. doi: 10.1186/1742-2094-10-151.
Point 4. We conducted densitometry analysis on the gel in IL-1β-treated astrocytes and results were given in line 173 the 928 bp fragment was expressed at a lower level (21%), whereas the majority of MCT4 mRNA (79%) was of normal size.
Point 5&6. Concerning Western blot analysis and other analyses, we hope to perform these experiments in the near future after obtaining a grant to cover the costs for a more thorough characterization. However, this manuscript focuses only the aberrant splicing and more specifically the intron retention during aging and AD pathology model in mice. We hope after this publication to have grants to continue this project and determine the levels of MCT-4 proteins in different mice brains (we already have the proteins) and in different lines with or without treatment. But it will be another study. At present, we unfortunately do not have time, no other funding to test MCT4 protein levels by WB.
Points 7. 8 and 9. We have now 4 WT mice and 4 AD of each age (12 WT and 12 AD) and not 8 as suggesting by this reviewer. It is a long, hard, and expensive work. I think it is enough to show the presence or not of transcripts normal or aberrant that it is our subject in this manuscript. We have not a single gel including all samples, but we added in supplementary figure other gels to confirm what we showed and presented. We presented in supplementary file 6 AD mice 3 month-old that we analysed, for the other types of mice only 4 were analysed. It is the reason why we declared 4 mice of each case.
Point 10. We added more details about the statistical analysis.
We added the number of independent cell cultures n=3 and a post-hoc test for Kruskall-Wallis test in 2.7. Statistical analysis.
‘All data for the differential expression of MCT4 constitutive transcript in endothelial cells with or without LPS & IL-β treatments (three independent cultures in each case) were analyzed using GraphPad Prism 5 (GraphPad Software, San Diego, CA). The findings were analyzed using one-way analysis of variance to detect differences between cells with LPS & IL-β treatments and cells without treatment as a control. A Kruskal-Wallis test with a Dunns multiple comparison test (a post-hoc test for Kruskal-Wallis test) was applied for multiple variable comparisons. The significance level was set at p < 0.05.
’
Minor comments: The minor comments were corrected
- In line 99, there is a typo: "0.3 x 106 cells/ml." The exponent "6" should be in superscript form (10^6). 106
- In line 50, there is a typographical error in the phrase "This the case in." we modify to ‘This is the case in’
We hope this modified version of our manuscript is obvious and eliminate all points of confusion and will allow acceptance of our manuscript.

Reviewer 2 Report
The authors reported that aged AD tg mice (APPswePS1dE9 showed a difference in MCT4 transcript. Sequencing results of MCT4 RT-PCR products displayed the presence of a remaining intron between exon 2 and 3, giving rise to the longer fragment detected by RT-PCR.
I think that further research is required to elucidate the mechanism of differential alternative splicing according to the aging and also to the cell types (astrocytes and pericytes). In addition, the effects of overexpression of longer form of MCT4 on behavioral phenotypes and astrocytic function in young AD mice is needed. However it is thought that this study is interesting and knowledgeable enough for readers in the present version.
I recomment a Minor revision.
1. In this study, the authors reported that MCT4 transcript was differentially expressed via alternative splicing according to the age in AD tg mice (APPswePS1dE9) as well as according to the cell types (in pericytes and astrocytes). Sequencing results of MCT4 RT-PCR products displayed the presence of a remaining intron between exon 2 and 3, giving rise to the longer fragment detected by RT-PCR.
2.It is thought that these results showing the differential expression of MCT4 via altanative splicing according to age and cell type is novel, interesting and knowledgeable for readers.
3.Based on the research of previous papers on MCT4, no other published data has studied two forms of MCT4, namely the shorter and longer forms of discriminatory expression via laternative splicing.
4.Further research is needed on differential alternative splicing mechanisms according to aging and different cell types (pericytes, astrocytes etc). In addition, the investigation on the effects of MCT4 overexpression of longer forms on the behavior phenotypic and astrocyte function in younger AD mice is needed.
5.The conclusions are consistent with the results and arguments presented in this study. In addition, the authors address the main question proposed.
6.I haven't done all the research, but I think it's usually presented correctly.
7. i)In Fig.1A and B and Fig.3, the information regarding the number of mice in each experimentla group is missing, it shoud be provided.
ii) The information on the the statistical analysis are insufficient and unclear. The description regarding statistical analysis should be provided more in detail.
Author Response
We highly appreciate the reviewers’ insightful and helpful comments on our manuscript.
Reviewer 2.
Firstly, I would thank you for these encouraging comments and the minor revisions that you have requested.
Secondly, we modified our manuscript to answer these two comments in point 7.
- We added the number of mice in Fig1A, 1B, and Fig3 as he (she) suggested in point 7(1) n=4.
- We added more details about the statistical analysis as he (she) suggested in point 7 (2).
We added the number of independent cell cultures n=3 and a post-hoc test for Kruskall-Wallis test in 2.7. Statistical analysis.
‘All data for the differential expression of MCT4 constitutive transcript in endothelial cells with or without LPS & IL-β treatments (three independent cultures in each case) were analyzed using GraphPad Prism 5 (GraphPad Software, San Diego, CA). The findings were analyzed using one-way analysis of variance to detect differences between cells with LPS & IL-β treatments and cells without treatment as a control. A Kruskal-Wallis test with a Dunns multiple comparison test (a post-hoc test for Kruskal-Wallis test) was applied for multiple variable comparisons. The significance level was set at p < 0.05.
’
We hope this modified version of our manuscript is obvious and eliminate all points of confusion and will allow acceptance of our manuscript.
Round 2
Reviewer 1 Report
The authors have effectively addressed my concerns.